# Field Trial with Vaccine Candidates Against Bovine Tuberculosis Among Likely Infected Cattle in a Natural Transmission Setting

**DOI:** 10.3390/vaccines12101173

**Published:** 2024-10-17

**Authors:** Ximena Ferrara Muñiz, Elizabeth García, Federico Carlos Blanco, Sergio Garbaccio, Carlos Garro, Martín Zumárraga, Odir Dellagostin, Marcos Trangoni, María Jimena Marfil, Maria Verónica Bianco, Alejandro Abdala, Javier Revelli, Maria Bergamasco, Adriana Soutullo, Rocío Marini, Rosana Valeria Rocha, Amorina Sánchez, Fabiana Bigi, Ana María Canal, María Emilia Eirin, Angel Adrián Cataldi

**Affiliations:** 1Instituto de Agrobiotecnología y Biología Molecular (IABiMo), UEDD CONICET-INTA, Centro de Investigación en Ciencias Veterinarias y Agronómicas (CICVyA)-CNIA, Hurlingham 1686, Buenos Aires Province, Argentina; 2Instituto de Patobiología Veterinaria (IPVet), UEDD CONICET-INTA, Instituto Nacional de Tecnología Agropecuaria (INTA), Hurlingham 1686, Buenos Aires Province, Argentina; 3Núcleo de Biotecnología, Centro de Desenvolvimento Tecnológico, Universidade Federal de Pelotas, Pelotas 96010-770, Río Grande do Soul, Brazil; 4Cátedra de Enfermedades Infecciosas, Facultad de Ciencias Veterinarias, Universidad de Buenos Aires, Ciudad Autónoma de Buenos Aires 1113, Argentina; 5Instituto Nacional de Tecnología Agropecuaria, Instituto de Fisiología y Recursos Genéticos Vegetales, Córdoba 5119, Córdoba Province, Argentina; 6Instituto Nacional de Tecnología Agropecuaria, Estación Experimental Agropecuaria Rafaela, Rafaela 2300, Santa Fe Province, Argentina; 7Veterinary Practitioner, Private Activity, San Martín 20, San Guillermo 2347, Santa Fe Province, Argentina; 8Laboratorio de Diagnóstico e Investigaciones Agropecuarias, Ministerio de Desarrollo Productivo de Santa Fe, Santa Fe 1251, Santa Fe Province, Argentina; 9Cátedra de Inmunología Básica, Facultad de Ciencias Bioquímicas y Biológicas, Universidad Nacional del Litoral, Santa Fe 3000, Santa Fe Province, Argentina; 10Cátedra de Patología Veterinaria, Facultad de Ciencias Veterinarias, Universidad Nacional del Litoral, Esperanza 3080, Santa Fe Province, Argentina

**Keywords:** tuberculosis, bovine, vaccine, live attenuated, field trial

## Abstract

**Background/Objectives**: Vaccines may improve the control and eradication of bovine tuberculosis. However, the evaluation of experimental candidates requires the assessment of the protection, excretion, transmission and biosafety. A natural transmission trial among likely infected animals was conducted. **Methods**: Seventy-four male heifers were randomly distributed (five groups) and vaccinated subcutaneously with attenuated strains (*M. bovis Δmce2* or *M. bovis Δmce2*-*phoP*), a recombinant *M. bovis* BCG Pasteur (BCGr) or *M. bovis* BCG Pasteur. Then, they cohoused with a naturally infected bTB cohort under field conditions exposed to the infection. **Results**: A 23% of transmission of wild-type strains was confirmed (non-vaccinated group). Strikingly, first vaccination did not induce immune response (caudal fold test and IFN-gamma release assay). However, after 74 days of exposure to bTB, animals were re-vaccinated. Although their sensitization increased throughout the trial, the vaccines did not confer significant protection, when compared to the non-vaccinated group, as demonstrated by pathology progression of lesions and confirmatory tools. Besides, the likelihood of acquiring the infection was similar in all groups compared to the non-vaccinated group (*p* > 0.076). Respiratory and digestive excretion of viable vaccine candidates was undetectable. To note, the group vaccinated with *M. bovis Δmce2-phoP* exhibited the highest proportion of animals without macroscopic lesions, compared to the one vaccinated with BCG, although this was not statistically supported. **Conclusions**: This highlights that further evaluation of these vaccines would not guarantee better protection. The limitations detected during the trial are discussed regarding the transmission rate of *M. bovis* wild-type, the imperfect test for studying sensitization, the need for a DIVA diagnosis and management conditions of the trials performed under routine husbandry conditions. Re-vaccination of likely infected bovines did not highlight a conclusive result, even suggesting a detrimental effect on those vaccinated with *M. bovis* BCG.

## 1. Introduction

Bovine tuberculosis (bTB) is a global animal disease caused by *Mycobacterium bovis* (*M. bovis*) and, to a lesser extent, other MTBC members, which affects cattle and other mammals [1]. This disease is detectable in 3% of bovines in Argentina and causes significant production losses primarily through weight loss in cattle (36%), reduced milk production (13%) and condemnation of carcasses at the slaughterhouses and abattoirs (10%). This represents an annual loss of USD 63,000,000 (Ministry of Agriculture, National Directorate of Animal Health, 1999). Positive reactor cows exhibited lactations of variable duration compared to the expected 305 days, as well as a decrease in daily production, and difficulties in reproduction, unlike non-reactive cows [2]. 

Despite the significant concern of bTB, regarding livestock health, its zoonotic nature and productivity problems, there is still no commercial vaccine available to control this disease. The only commercial vaccine available worldwide is *M. bovis* Bacillus Calmette Guérin (BCG), which is used to control human tuberculosis, and protects against disseminated and meningeal tuberculosis in children [3,4].

Previous experimental trials in cattle, mainly using calves, have evaluated the protective efficacy of different *M. bovis* BCG strains (*M. bovis* BCG Tokyo, *M. bovis* BCG Danish, *M. bovis* BCG Russia) as vaccine candidates, under bTB natural transmission settings. These trials differed in the methods used to evaluate protection and thus, in the magnitude of protection obtained among vaccinated calves, which ranged from 22.4% to 86.7% [5,6,7,8,9,10]. 

Despite the extensive knowledge generated around the use of different *M. bovis* BCG strains as a candidate vaccine in cattle, no eradication campaign includes any of these strains. Bovine vaccination can interfere with official diagnostic methods that are based on the use of the bovine protein purified derivative (PPDB) as a diagnostic antigen, either in the Tuberculin Skin Test (TST) or in the Interferon Gamma Release Assay (IGRA). However, in wildlife, *M. bovis* BCG has been used as a vaccine in Britain since 2010 and as an experimental vaccine in New Zealand [8]. 

Considering the sensitization of *M. bovis* BCG-vaccinated animals, it decreased from 80% to 8% after six to nine months, as investigated by the Single Cervical TST (SCTST) [11]. In addition, Nuggent et al. [8,9] reported that *M. bovis* BCG vaccination did not significantly affect the response to the TST or IGRA seven months after vaccination [8,9]. 

To improve *M. bovis* BCG protection, Rizzi et al. [12] developed a genetically modified *M. bovis* BCG Pasteur vaccine candidate that over-expresses the Ag85B protein, an immunodominant antigen with an essential role in pathogenesis. This candidate, named *M. bovis* ΔleuD BCG-85B, protected cattle better than the wild-type *M. bovis* BCG Pasteur under experimental conditions [12].

Moreover, specific antigens such as ESAT-6 and CFP-10 (present in *M. bovis* wild-type strains but absent in *M. bovis* BCG), or Mb3645c/Rv3615c (secreted in *M. bovis* wild-type strains but not in *M. bovis* BCG) could replace PPDB [13] to differentiate infected from vaccinated animals (DIVA antigens) [14]. For instance, IGRA testing performed using the recombinant antigens ESAT6-CFP10 has shown the ability to differentiate unprotected vaccinated from protected vaccinated animals [6,7]. 

Despite the use of the *M. bovis* BCG vaccine, the generation of new potential *M. bovis* candidates may be useful as an alternative to improve the protection of *M. bovis* BCG and to allow for complementary DIVA diagnosis. In this regard, a functional genomic approach characterized by gene knock-out mutants with different levels of attenuation has led to the concept that rationally attenuating live and replicating mutants of *M. tuberculosis*/*M. bovis* are potential vaccine candidates against tuberculosis [15,16,17,18,19].

In Argentina, the *M. bovis* NCTC10772 strain, with deletions in the *mce2A* and *mce2B* genes, gave rise to an experimental strain named *M. bovis Δmce2*. Animals vaccinated with this experimental strain and then challenged with a pathogenic *M. bovis* strain under controlled conditions remained alive for 100 days. Upon necropsy, the *M. bovis Δmce2*-vaccinated group displayed a lower macro and microscopic score compared to the group vaccinated with *M. bovis* BCG Pasteur or to the non-vaccinated group [17,20]. 

Subsequently, a candidate *M. bovis* strain was developed, derived from an additional mutation in the *phoP* gene of the *M. bovis Δmce2* strain, which was found to be significantly more attenuated [19].

Despite these previous experimental controlled studies, there is no previous research regarding their performance as vaccine candidates in a natural transmission setting. In the present study, the use of *M. bovis Δmce2*, *M. bovis Δmce2*-*phoP* and *M. bovis* ΔleuD BCG-85B strains was assessed as vaccine candidates in cattle focusing on the immune response, the protection in terms of the reduction in pathogenicity and on the biosafety conferred under a natural transmission setting. 

## 2. Materials and Methods

### 2.1. Conception of the Study

The bovine model is the natural host of *M. bovis* and, therefore, the bovine should be the model used to evaluate vaccine candidates. According to the literature [5,6,7,8,9] and previous experiences with the experimental strains evaluated in this study [18], the trial consisted of a randomized design with a period of pre-vaccination, vaccination, and cohousing of vaccinated and control groups with infected cattle (in a one-to-one ratio) for 433 days in a pen. Figure 1 describes the timeline of the trial: from its beginning in November 2018 until March 2020. It should be noted that a re-vaccination was carried out because of an apparent lack of significant immune response after the first vaccination. The Caudal Fold Test (CFT) and IGRA, as correlates of the cell-mediated immune response, were the methods used to monitor the sensitization of animals to the vaccine strains in the pre-cohousing period, and afterward, during the coexistence phase of the study. Sensitization could also occur during the coexisting phase because of the contact with naturally infected cattle. Serology for bTB and *M. avium* subsp. *paratuberculosis* (MAP) was also performed to monitor the existence of false-negative and false-positive cattle, respectively. Furthermore, stool samples and nasal swabs were taken to detect possible excretion of mycobacteria through the digestive and respiratory routes, respectively. Water and soil samples were also analyzed. At the end of the trial, a detailed necropsy was performed for all animals.

### 2.2. Studied Animals

The trial was conducted in a productive dairy region of the country, on a farm located in Colonia Rosa, San Cristobal Department, Santa Fe province (lat. 30°18′00″ S, long. 61°58′59″ W), Argentina. At the time of the study, the farm was under a sanitation program. According to the SCTST, 80 animals were identified as reactors. These animals were housed in a pen and maintained in semi-extensive conditions. 

On the other hand, 74 three-month-old male calves of the Holando-Argentino breed were selected from a dairy farm with at least 5 years of *M. bovis*-free herd history, located in an experimental field of the National Institute of Agricultural Technology, in Rafaela city, Santa Fe province (lat. 31°16′00″ S, long. 61°29′00″ W), Argentina. The selection criteria included being SCTST-negative, according to the National Program for the Control and Eradication of bTB (Senasa, National Service for Health and Food Quality) [20] (Resolution 128/2012). All selected animals tested negative for MAP serological diagnosis. These animals were randomly distributed in five different groups and named according to the strain used for vaccination: *M. bovis* Δ*mce2* (n = 15), *M. bovis Δmce2-phoP* (n = 15), *M. bovis* BCGr (n = 15), *M. bovis* BCG (n = 14) and the non-vaccinated group (n = 15). 

### 2.3. Bacterial Strains, Inoculum Preparation and Vaccination

The following *M. bovis* strains were used for the present trial: the vaccine candidates, *M. bovis ∆mce2* [18], *M. bovis ∆mce2-phoP* [19] and *M. bovis* ∆leuD BCG-85B (BCGr) [12]. Additionally, the *M. bovis* BCG Pasteur 1173P2 strain was included as a positive control and Phosphate Buffered Saline (PBS) 1X was inoculated as negative control. 

The viability of the bacteria was monitored using a commercial kit (Live/Dead BacLight™ Bacterial Viability kit, Invitrogen Molecular Probes, Carlsbad, CA, USA) following the manufacturer’s instructions. Inocula were prepared from those cultures that exhibited ≥90% viability. The concentration of the inoculum was determined considering an OD 600 nm of 0.1 was equivalent to a titer of 1 × 10^6^ CFU/mL.

Vaccination was performed subcutaneously on the side of the neck with a 2 mL suspension in PBS 1X containing 1 × 10^6^ CFU of *M. bovis Δmce2*, *M. bovis Δmce2-phoP*, *M. bovis* BCGr or *M. bovis* BCG. In the case of the non-vaccinated group, the inoculum consisted of 2 mL of sterile PBS 1X. Subsequently, the animals remained isolated for 3 months in a pen where CFT-positive cattle had never been present before. 

The vaccinated and non-vaccinated groups were then mixed with an equal number of *M. bovis* naturally infected bovines in a pen measuring 75 m long × 62 m wide. Re-vaccination was performed at 164 days post-vaccination (dpv), with 5 × 10^7^ CFU, following the same protocol described for the initial vaccination. This change was introduced in the original protocol, which only included one vaccination time, because the cell-mediated immune response was lower than expected at 75 dpv. This poor sensitization included one animal that was positive for IGRA, that died during the trial (for this reason these data were not considered for the analysis) and three others that were CFT positive reactors. All these four animals belonged to the *M. bovis ∆mce2* group. Thus, zero time was considered from the re-vaccination (Figure 1).

### 2.4. Caudal Fold Test

CFT was carried out by a veterinarian accredited by the National Service for Health and Food Quality. The test was performed using 0.1 mL of PPDB (Bovine tuberculin PPD 3000, Prionics, Lelystad, The Netherlands) according to the current regulation (Resolution 128/2012). During the trial, the CFT was performed at 90 dpv and at 0, 84 and 237 days’ post re-vaccination (dprv).

### 2.5. Interferon-Gamma Release Assay

Heparinized blood samples were dispensed in 200 µL aliquots into individual wells of a 96-well plate (Biofil, AP Biotech, Buenos Aires, Argentina). Wells containing whole blood were individually stimulated with 25 μL of commercial PPDB (Bovine tuberculin PPD 3000, Prionics, Lelystad, The Netherlands) and with avian protein purified derivative (PPDA) (Avian tuberculin PPD 2500, Prionics, Lelystad, The Netherlands) at a final concentration of 10 μg/mL. Additionally, a histidine-tagged fusion recombinant protein of ESAT-6, CFP-10, and Rv3615c (FP) (final concentration of 10 μg/mL) was used for the assay [14]. Pokeweed mitogen (PKM) (Bovigam^®^ Pokeweed Mitogen, Prionics, Schlieren, Switzerland) was used at a final concentration of 5 μg/mL, as a control of T cell viability, and PBS 1X was used as a nil. Blood cultures were incubated at 37 °C in a humid atmosphere with 5% CO_2_ for 18 h. Plasma was harvested and stored at −20 °C. IFN-gamma release in stimulated plasma was determined using a commercial enzyme-linked immunosorbent assay (ELISA) (Bovigam^®^ TB Kit; Thermo Fisher, Buenos Aires, Argentina) according to the manufacturer’s instructions. The test was performed for the screening of the naïve animals to be vaccinated, and at four times during the trial at 75 dpv and at 42, 144 and 249 dprv.

The measurement of the OD 450 nm value with a reference filter of 620 nm was carried out using a spectrophotometer equipment (Multiskan^®^ Spectrum, Thermo, Vantaa, Finland) applying the reading criterion described by the manufacturer. Standard criteria were used as follows: an animal was considered positive when the difference between OD PPDA and OD PBS 1X was greater than or equal (≥) to 0.1 (OD PPDA—OD PBS 1X ≥ 0.1) and, simultaneously, when the difference between OD PPDA and OD PPDA was also greater than or equal to 0.1 (OD PPDB—OD PPDA ≥ 0.1), indicating infection with *M. bovis*. Those animals whose OD determinations did not comply with this criterion were considered negative. In the case of the FP, animals were considered positive when the difference between OD FP and OD PBS was greater than or equal to 0.1 (OD FP—OD PBS ≥ 0.1).

### 2.6. Serology

This test was performed to evaluate the humoral immune response by detecting antibodies against *M. bovis* and MAP, which is also endemic in Argentina, and a possible cause of false-positive animals. For MAP detection, an indirect ELISA was performed using the commercial antigen PPDA (Allied Monitor Inc., Fayette, MO, USA). Plasma from the animals was collected at two months’ pre-vaccination (mpv) from naïve animals and at 75 dpv. Animals whose OD 405 nm exceeded 50% of the value compared to the positive control were considered positive. The procedures and details of the technique have been previously described by Moyano et al. [21]. The presence of anti-*M. bovis* antibodies were assessed with an indirect ELISA test using the PPDB antigen (Bovine tuberculin PPD 3000, Prionics, Lelystad, The Netherlands). Plasma from the animals was collected at 75 dpv, 42 and 249 dprv. The procedures and details of the technique have been previously described by Garbaccio et al. [22].

### 2.7. Slaughterhouse Inspection and Sample Collection

After 266 dprv, the animals were moved to a commercial abattoir located 240 km from the dairy farm of the trial. A detailed post mortem inspection was performed in the slaughterhouse, following procedures previously described [23]. Necropsy was conducted focusing on the presence of lesions compatible with bTB (LCTs). Lungs (L), pulmonary lymph nodes (LNs), including the tracheobronchial and mediastinal lymph nodes, liver and lymph nodes of the cranial region (submandibular and retropharyngeal LNs), and the digestive system (mesenteric LNs), were systematically examined. LCTs were registered and converted to scores according to Garbaccio et al. depending on location in the Ls or LNs, the number of lesions, size, red halo, presence of capsule, color, calcification, and percentage of lesion surface area affected [23]. For bacteriology and tissue-PCR, a ~5 cm × 5 cm sample of each of the organs was collected in duplicate and placed in sterilized jars. The ones intended for bacteriological analysis were transported refrigerated and then stored at −20 °C until processing. For histopathological analysis, 0.3 cm × 0.5 cm thick LNs and L samples were placed in cassettes and introduced in 10% buffered formalin. When LCTs were detected in the liver or spleen, 1 cm × 4 cm samples were collected. 

### 2.8. Bacteriology and Histopathology

Each tissue sample was mechanically macerated for 3 min using a Stomacher (Masticator Basic IUL Instruments Model No. 470, Barcelona, Spain). Then, 4 mL of the homogenate were taken to carry out decontamination through the Petroff method (NaOH 4%) to eliminate the associated microbiota. The product obtained was plated in triplicate in Stonebrink culture media, and incubated at 37 °C. The plates were checked on a weekly basis (for 60 days) to await colony growth. 

Samples were taken from the pen and water troughs where the animals were kept to assess the possible presence of *M. bovis* and Non-Tuberculous Mycobacteria (NTM) in soil and water. Bacteriological isolation was performed on Stonebrink and on Herrold media supplemented with mycobactin, as described by Tortone et al. [24]. In turn, samples were cultured again by supplementing the culture medium with PANTA (Supplement antimicrobial containing Polymyxin B, Amphotericin B, Nalidixic Acid, Trimethoprim, and Azlocillin). 

If mycobacterial growth developed, samples were stained with Ziehl-Neelsen (ZN) to detect Acid-fast bacilli (AFB), as previously described [25]. Lungs and LNs (with or without LCTs) were collected, fixed in 10% buffered formalin, sectioned, and embedded in paraffin wax. Five-millimeter-thick sections were stained with Hematoxylin-Eosin and ZN for further histopathological examination, as previously described [25]. 

Histological sections were observed under a microscope (Nikon 80i with Nikon DS-Fi1c camera). A microscopic score was developed according to Wangoo et al. [26] with variations modified by Ana Canal (unpublished data), considering the presence of necrosis, calcification, Langhans giant cells, fibrosis, and predominance of mononuclear or polymorphonuclear cells. The microscopic score (grades I, II, III, and IV) of each animal was obtained by adding the number of granulomas present in L and LN sections of the bovines for each group, as described by Wangoo et al. [26].

### 2.9. Templates Preparation

DNA extraction was carried out from tissue samples (Ls and LNs) using a commercial kit (ADN PuriPrep-T kit, Inbio Highway, Tandil, Argentina) following the manufacturer’s instructions. Then, the genetic material was quantified by spectrophotometry (NanoDrop™, Thermo Fisher Scientific, Buenos Aires, Argentina). Nasal swabs were suspended in 5 mL of sterile PBS 1X. The swab was wrung out to extract the contents of the bacilli presumptively present, and a pellet was obtained after centrifugation at 5000 rpm for 15 min. The swabs pellets were pooled in groups of five samples (including the same five animals in each sampling) from which DNA was extracted using a commercial extraction kit (ADN PuriPrep T kit, Buenos Aires, Argentina) following the manufacturer’s instructions. 

DNA extraction from feces was performed from 0.2 g of pooled sample belonging to five animals from the same study group, using a commercial kit (QIAamp PowerFecal DNA Kit, Qiagen, Buenos Aires, Argentina) and following the manufacturer’s instructions. 

DNA from mycobacterial isolates was obtained by thermal lysis. Colonies were suspended in 300 µL of sterile water, heated at 95 °C for 40 min and centrifuged at 12,000 rpm for 10 min. Then, 5 µL of the supernatant was used as a template for the PCR. The analysis of NTM was performed as described for mycobacterial isolates.

### 2.10. Polymerase Chain Reaction

A variety of PCRs were used to identify strains involved in the lesions, which contributed to the knowledge of the transmissibility and safety of the strains under study (Appendix A). DNA samples extracted from bovine tissues or from isolates obtained from tissues (colony-PCR) were used as templates. Additionally, respiratory and digestive excretions were analyzed in nasal swabs and feces, respectively. Isolates from water and soil samples present in the pen during the trial were analyzed (colony-PCR) to obtain information on the possible excretion of *M. bovis* strains into the environment. Furthermore, these environmental isolates were tested to identify NTM species.

β actin-PCR [27] was performed to check the integrity of the template. In cases with no amplification, another sample was extracted to perform a new evaluation. The PCR algorithm was only proceeded with when a positive result was obtained. A positive result for the IS*6110*-PCR [28] or Rv*2807*-PCR [29] indicated the presence of *M. bovis* in the tissue, which prompted the differential identification of the experimental *M. bovis* strains, the wild-type *M. bovis* strains and the *M. bovis* BCG strains. Subsequently, the Mut. *Mce2* del-PCR [17] allowed us to discriminate between the presence or absence of the *mce2A-B* operon, which was absent in the experimental vaccine strains but present in the *M. bovis* wild-type, the *M. bovis* BCG and *M. bovis* BCGr strains. The *phoP*-PCR [19] discriminated between *M. bovis ∆mce2* strain (presence of the *phoP* gene) and *M. bovis ∆mce2-phoP* (*phoP* gene deletion) strain. Finally, *esxA*-PCR or *esxB*-PCR [30] differentially identified *M. bovis* wild-type strains from *M. bovis* BCG and *M. bovis* BCGr strains. This is possible because the genes *esxA* (ESAT-6) and *esxB* (CFP-10) are present in the RD1 region, a region absent from the *M. bovis* BCG and *M. bovis* BCGr strains but present in *M. bovis* wild-type strains (Table 1). 

NTM species were identified by amplifying the *16S* subunit of Ribosomal RNA (*16S* rRNA), the gene encoding the heat shock protein (*hsp65*), and the gene that encodes the β subunit of RNA polymerase (*rpoB*), as described by Kirschner and Böttger [31], Telenti et al. [32] and Adékambi et al. [33], respectively.

Validation of the PCR reaction was performed using DNA from the reference strain *M. bovis* AN5, *M. bovis* NCTC 10772 and *M. bovis* BCG Pasteur as well as DNA from the experimental vaccines *M. bovis Δmce2* and *M. bovis Δmce2-phoP* as positive controls. Furthermore, DNA extracted from tissue of a previously characterized animal with bacteriological confirmation and a negative contamination control (DNase-free water) were included in the analysis. When the template was obtained from tissue, a negative extraction control was incorporated, which was processed in parallel with each round of tissue extraction performed with the commercial kit.

The PCR products were visualized by electrophoresis on 2% agarose gels (Trans, AP Biotech, Buenos Aires, Argentina) in TAE 1X buffer, stained with 5 mg/mL of ethidium bromide (Promega, Madison, WI, USA), and visualized by ultraviolet light. The presence or absence of the expected size was the criterion used for detecting *M. bovis* strains. 

In the case of NTM, amplicons were purified using the Illustra DNA and Gel Band Purification Kit (GE Healthcare, Buckinghamshire, UK). The sequences were obtained through automatic sequencing spanning double frameshifts and compared with sequences from the database online from the National Centre for Biotechnology Information (https://blast.ncbi.nlm.nih.gov/Blast.cgi, accessed on 10 January 2024). For the *hsp65* gene and the *16S* rRNA sequences, the comparisons were performed with Gene BLAST (http://hsp65blast.phsa.ca/blast/explosión.html, accessed on 10 January 2024) and the Ribosomal Database Project (http://rdp.cme.msu.edu, accessed on 10 January 2024), respectively. The identification criterion of the NTM was obtained through a consensus among the identities provided by at least two of the three genes studied [33,34].

### 2.11. Spoligotyping

Molecular typing was performed as previously described by Kamerbeek et al. (1997), using a commercial kit (MapmyGenome™, Hyderabad, India). The spoligotypes obtained were compared with those present in a local database of the IABIMo INTA-CONICET, and with an international database (https://www.mbovis.org/, accessed on 20 January 2024) managed by VISAVET Health Surveillance Centre of Universidad Complutense, Madrid. *M. tuberculosis* H37Rv (ATCC 27294) and *M. bovis* BCG (ATCC 27289) were included as reference strains.

### 2.12. Transmission Rate

The transmission rate was calculated as the percentage of animals in the non-vaccinated group with evidence of sensitization (CFT and IGRA) and confirmation of *M. bovis* infection (LCT, histopathology, tissue-PCR and bacteriology followed by colony-PCR). 

For the vaccinated groups, the likelihood of acquiring the disease through natural transmission of *M. bovis* wild-type strains was analyzed considering direct diagnosis (histopathology, tissue PCR and bacteriology). In this study, immunological indices were not considered, since the sensitization detected ante mortem could have been associated with vaccination or infection.

### 2.13. Statistical Analysis

We will refer to the total number of animals that survived until the end of the trial (67 bovines) due to the feasibility of performing ante mortem diagnosis to confirm the disease. These animals included 14 bovines in each of the following groups: *M bovis ∆mce2, M. bovis ∆mce2-phoP* and *M. bovis* BCGr; 13 bovines in the non-vaccinated group and 12 bovines in the *M. bovis* BCG group. 

Data distribution was tested by the Shapiro–Wilk test (GraphPad Software, Inc., San Diego, CA, USA). The prevalence considered as the percentage of the positive animals for a diagnostic tool over the total animals was recorded for CFT, IGRA and the macroscopic and microscopic scores. The incidence of the positivity for the CFT and IGRA was represented using the Kaplan–Meier plot with the log-rank test. The frequency of positive cattle, represented as percentage (LCT, bacteriology, tissue-PCR or histopathology), was calculated with its proper 95% confidence interval and compared using the “comparison method by proportions”; both were calculated with EPIDAT Version 3.1. Significant differences were considered to be found when the *p* value was <0.05. The statistical analysis of the macroscopic and microscopic scores was the same as that used for IGRA and CFT. Statistics and graphs were performed using the GraphPad Prism Program version 6.04 for Windows (GraphPad Software, USA).

Logistic regression was performed to evaluate the probability of infection between the different vaccinated groups compared to the non-vaccinated group. For this purpose, the *glm* function of the *nlme* package (https://cran.r-project.org/web/packages/nlme/nlme.pdf, accessed on 4 February 2024) was used under the R environment (R Core Team. 2024. Version 4.4.0).

## 3. Results

### 3.1. Cell-Mediated Immune Response of Vaccinated Cattle

The total positive CFT reactors in the experimental groups after re-vaccination and exposure to naturally *M. bovis* infected cattle was 38.8% (25.3–56.9). These experimental CFT reactors were distributed in decreasing proportion as follows: 78.6% (49.2–95.3) in the *M. bovis ∆mce2* group, 35.7% (12.8–64.9) in the *M. bovis* ∆*mce2-phoP* group, 33.3% (9.9–65.1) in the *M. bovis* BCG group, 23% (5.0–53.8) in the non-vaccinated group and 21.4% (4.7–50.8) in the *M. bovis* BCGr group. 

Results related to the original vaccination showed a lack of reactivity, except for the *M. bovis ∆mce2* group. The reactivity discriminated by the stimulation antigen in the IGRA test was null at 75 dpv (Figure 2B), with a low CFT reactivity (21.4%; 3/14) (Figure 2A) at 90 dpv. After the re-vaccination, although the reactivity varied, the trend was positive throughout the trial until 84 dprv, as the percentage of CFT positives increased in all groups. In animals vaccinated with *M. bovis ∆mce2* and *M. bovis ∆mce2-phoP*, these CFT percentages continued to increase until 247 dprv, but without significant differences compared to the other groups (*p* > 0.05) (Figure 2A). 

When stimulation with PPDs was performed in the IGRA, 70.1% (51.5–93.2) of the bovines were positive reactors at least once during the trial. The groups vaccinated with *M. bovis* BCG (50%; 21.1–78.9) and *M. bovis* BCGr (42.8%; 17.7–71.1), exhibited the lowest proportion of positive results, followed by the non-vaccinated group (76.9%; 46.2–94.9). The highest proportion of positive animals to the IGRA corresponded to *M. bovis ∆mce2* (100%; 69.1–100) and *M. bovis ∆mce2*-*phoP* (92.8%; 66.9–99.8) groups, although the differences among all the groups were not significant (*p* > 0.05) (Figure 2B). 

A comparison of the cell-mediated immune tests used in this study confirmed that IGRA detected a greater number of animals than the CFT (Figure 2C,D).

The magnitude of the IGRA response (measured by OD values) varied between groups throughout the trial, although without significant differences (*p* > 0.05). Additionally, stimulation with PPDA antigen, an indicative of NTM sensitization, yielded significantly lower values than stimulation with PPDB across all groups (*p* < 0.05) (Figure 3A). When stimulated with PPDB, a variable magnitude of response was observed among groups and in turn it varied significantly in each group at the different sampling points, especially in the groups vaccinated with *M. bovis Δmce2* and *M. bovis Δmce2*-*phoP* (Figure 3B). In the case of stimulation with the FP, something similar happened, but the increase was more significant towards the end of the trial (Figure 3C).

### 3.2. Humoral Response

The serological response against *M. bovis* showed weak reactivity, and the ELISA only detected three positive animals in the final stage of the trial (249 dprv). These positive animals were from the non-vaccinated, *M. bovis* BCGr, and *M. bovis Δmce2* groups. Additionally, no positive animals were identified for anti-MAP antibodies throughout the trial.

### 3.3. Excretion Study from Biological and Environmental Sources

Respiratory excretion of mycobacteria, evidenced via nasal swabs, increased in all the groups by the end of the trial. The *M. bovis Δmce2*-*phoP* group had the highest number of PCR-positive animals (64.3%; 35.1–87.2), followed by the *M. bovis* BCG (58.3%; 27.7–84.8), *M. bovis Δmce2* (57.1%; 28.9–82.3), the non-vaccinated (53.8%; 25.1–80.8), and *M. bovis* BCGr groups (42.9%; 17.7–71.1).

Regarding the fecal samples, no positive animals were detectable for either PCR-IS*6110* or PCR-Rv*2807* in any of the groups. The bacteriological analysis of the surrounding environment revealed the presence of a fast-growing, chromogenic isolate identified as *Nocardia* sp. but no *M. bovis* strains. 

### 3.4. Mortality Detected During the Trial

As the trial progressed, 9.5% (2.1–16.8) of the calves died. Presumptive diagnosis was performed in all cases. A 13.3% (1.7–40.5) mortality was detected in the non-vaccinated group (chronic pneumonia at 110 dpv and Bovine Viral Disease/Hematophagous parasites at 160 dpv); 14.3% (1.8–42.8) in the *M. bovis* BCG group (Hemorrhagic enteritis at 22 dpv and reticulopericarditis at 103 dpv); a 6.7% (0.2–31.9) in the *M. bovis* BCGr group (stuck in the feeder at 142 dpv), a 6.7% (0.2–31.9) in the *M. bovis ∆mce2* group (stuck in the feeder at 197 dprv) and a 6.7% (0.2–31.9) in the *M. bovis ∆mce2-phoP* group (liver abscess, LN precrural with caseous contents at 236 dprv).

### 3.5. Slaughterhouse Inspection and Confirmatory Diagnosis

Regarding the macroscopic lesions, 20.9% (14/67) of the animals exhibited LCT. The *M. bovis* BCG (41.7%; 15.2–72.3) and *M. bovis* BCGr (21%; 4.7–50.8) groups had the highest proportion of LCT, followed by the groups vaccinated with the *M. bovis Δmce2* (21%; 4.7–50.8) and *M. bovis Δmce2-phoP* (7.1%; 0.2–33.9) strains. Finally, the non-vaccinated group, had the lowest number of animals with LCT (15.3%; 1.9–45.5). The degree of LCT progression revealed that the groups vaccinated with *M. bovis* BCG and *M. bovis* BCGr had the highest score of LCT, followed by the groups vaccinated with *M. bovis Δmce2* and *M. bovis Δmce2-phoP*. The non-vaccinated group had the lowest score. Despite these variations, the differences of LCT proportions and disease progression among groups were not significant (*p* > 0.05) (Figure 4A). 

Considering the microscopic lesions, 20.9% (10.4–31.4) of the different groups had histopathological lesions, with a predominance of lesions in the respiratory system and LNs. The lesions were most frequently in the LNs of the cranial region, mediastinal LNs and L. The group vaccinated with *M. bovis* BCG had the highest number of animals with microscopic lesions (41.7%; 15.2–72.3), followed by the non-vaccinated group (23%; 5.0–53.8), *M. bovis* BCGr (21.4%; 4.7–50.8), *M. bovis Δmce2* (14.3%; 1.8–42.8) and *M. bovis Δmce2*-*phoP* (7.1%; 0.2–33.9). These proportions did not vary significantly (*p* ≥ 0.11) (Figure 4B). 

Table 2 displays a detailed comparison of macroscopic and microscopic scores. 

Tissue-PCR revealed the presence of the *M. bovis* wild-type genome in 16 bovines, and the lack of genomic DNA from experimental vaccine candidates. It is noteworthy that in the groups of animals vaccinated with *M. bovis* Δ*mce2, M. bovis* Δ*mce2-phoP*, and *M. bovis* BCGr strains, two animals without lesions exhibited positive PCR results. In the case of the group vaccinated with *M. bovis* BCG, one animal lacked macro or microscopic lesions. The results obtained by bacteriology also revealed the presence of *M. bovis* wild-type strains (AFB and positive by colony-PCR) in six animals from the *M. bovis Δmce2-phoP*, *M. bovis Δmce2*, *M. bovis* BCGr, and non-vaccinated groups (one each). Two isolates from the *M. bovis* BCG group were obtained (n = 2). 

The detection of spoligotype SB013 reinforced the sole presence of *M. bovis* wild-type isolates among the animals included in the trial, as this spoligotype is not associated with either *M. bovis* BCG or *M. bovis* NCTC10772, the parental strain of *M. bovis Δmce2* and *M. bovis Δmce2*-*phoP*.

### 3.6. Transmission Rate and Correlate of Protection

The transmission rate based on the CFT and IGRA was 23.1% (5.04–53.81) and 76.9% (46.19–94.96), respectively. These results represented the non-vaccinated animals that were subsequently sensitized during the trial. The use of confirmatory tools gave variable transmission rate: 15.3% (1.9–45.4) for LCT; 23.1% (5–53.8) for histopathology, 15.4% (1.9–45.4) for tissue-PCR-IS*6110* and 7.7% (0.2–36) for bacteriology. The combination of the confirmatory tools (bacteriology, histopathology and tissue-PCR) represented a transmission rate of 23.1% (5.04–53.81). Regarding vaccinated cattle exposed to *M. bovis* wild-type infection, the likelihood of acquiring the infection compared to the non-vaccinated group decreased as follows: *M. bovis* BCG (*p* = 0.08), *M. bovis Δmce2* (*p* = 0.24) and *M. bovis* BCGr (*p* = 0.24) and *M. bovis Δmce2-phoP* (*p* = 0.69). 

## 4. Discussion

A large, longitudinal study was performed using a natural transmission model in which *M. bovis* infected animals were in contact with animals vaccinated with a range of vaccines, including *M. bovis* BCG. The employed field conditions are useful to demonstrate the potential results of their use under real-world conditions. However, they have limitations. In the present study, it was observed that original vaccination did not induce immune response. On the one hand, we determined the vaccine viability and found that it was significantly reduced. It is likely that extreme high temperatures reported in the northern part of the country negatively affected the bacterial viability during those days. Cultures with ≥90% viability were used to prepare inocula; thus, this issue could not represent the limitation associated with the viability of mycobacteria. However, the inocula were transported from the lab to the field where the trial was performed, travelling for 800 km at room temperature following previous experiences [18]. During those days, an unusual heat wave occurred. We propose that viability was lost over time between the preparation of the strains and their inoculation in the animals. Previous studies showed a rapid beginning of the degradation of lyophilized/dried *M. bovis* BCG at 30–37°C, with an evident rate of count plate reduction, compromising the thermostability of the vaccine strain [35]. 

In the present trial, re-vaccination was performed in winter, with lower temperatures than at the time of vaccination. All protocols performed at this second inoculation time were the same, but an increase in the cell-mediated immune response was then observed (Figure 3 and Figure 4). To highlight, at this point the observed cell-mediated immune response could be partially attributed to vaccination, as animals had been exposed to naturally infected cattle for more than 70 days at the time of re-vaccination. This could not be confirmed because a DIVA diagnosis strategy was not available for non-BCG strains and, for *M. bovis* BCG strains, there were no intermediate sampling points that showed IGRA results for the fusion protein stimulus. 

To note, Jones et al., 2016, reported that vaccination with a heat-inactivated *M. bovis* strain through the intramuscular route induced IGRA and SCTST positivity among calves [36]. This study is in contrast to our hypothesis. However, the authors do not specify data on heating conditions, such as temperature, time of exposure, among others. It is possible that, in the present study, mycobacteria were subjected to temperatures above 37 °C for a prolonged time (hours), and then, this might have induced a considerable degradation rate that could have affected the immunogenicity associated with the antigens of the killed *M. bovis* strains.

Immune response induced by original vaccination was carried out before exposing the animals to the naturally infected TB cohort. CFT and IGRA testing performed at 75 and 90 days’ post original vaccination showed an extremely low reactivity rate, which was probably maintained until the cohousing with naturally infected bovines, and therefore vaccinated bovines were probably not immunized at this time. Considering this fact, re-vaccination, performed after more than 70 days of cohousing, could have been administrated in cattle likely infected. Undoubtedly, this situation prevented us from drawing conclusions about the protection exclusively conferred by re-vaccination. However, this trial shows results about the impact of vaccinating cattle that are may be already infected, a scenario that could occur if large-scale vaccination will be implemented in the livestock industry. This kind of trials could provide evidence about the incidence of progressive lesions with decreasing losses due to the condemnation of infected carcasses at the abattoir. Berggren and coworkers carried out a larger field vaccination trial among likely infected bovines. They concluded that the vaccination with *M. bovis* BCG had no effect on preventing bTB or diminishing the rate of condemnations at the slaughterhouse [35]. Consistent with this previous study, in the present trial vaccination did not improve protection associated with the progressive infection, based on macroscopic or microscopic lesions. Despite no statistical differences regarding LCT and histopathology among groups, the *M. bovis Δmce2-phoP*-vaccinated group had only one bovine with LCT and microscopic lesions, and therefore, the lowest proportion of affected animals (7.1%; 0.4–36.9), compared with the remaining groups. This finding may be explained by the possible infection of this animal prior to the re-vaccination time, supported by the high degree of progression of the lesions, and thus compatible with a longer time of infection. The remaining 13 animals vaccinated with *M. bovis Δmce2-phoP* only exhibited positive tissue-PCR in two animals in the respiratory system (mediastinal LNs and L). Might this be attributed to the protection conferred by re-vaccination or to the low transmission rate observed in the present study? Although we cannot confirm that this positivity is due to the presence of viable mycobacteria, these results also highlight the importance of including complementary diagnostic tools in trials investigating new vaccine candidates to allow a deeper comprehension of the infection status.

The present trial evaluates vaccines that may improve the protection conferred by *M. bovis* BCG in cattle. Currently, there are not many alternatives to these vaccine candidates under study, so the results are relevant, especially since field conditions have been applied to draw conclusions. Most studies reported in the literature have used different strains of *M. bovis* BCG and have demonstrated variable reductions in lesions in vaccinated cattle [5,6,7,37,38]. In contrast to these studies, in this work, the *M. bovis* BCG-vaccinated group was the one that presented the highest proportion (41.7%) of animals with LCT, in opposite to the non-vaccinated group (15.3%). These results are striking because vaccination with *M. bovis* BCG was expected to confer immune protection against natural exposure to *M. bovis* wild-type strains. As mentioned before, there is evidence supporting the failure of vaccination, however, the fact that the *M. bovis*-vaccinated group and the non-vaccinated one showed opposite results to those expected could suggest that re-vaccination induced a detrimental effect. Buddle and coworkers performed a trial with calves vaccinated with *M. bovis* BCG. They observed that re-vaccination of calves 6 weeks after the initial vaccination (at birth), resulted in reduced protection compared to a group vaccinated with a single dose [39], while re-vaccination of 5–6-month-old calves with BCG did not have a detrimental effect [40]. Despite this result, in the present study, the re-vaccinated animals were older (nine-month-old) than the calves reported by Buddle and coworkers and closer to the animals reported by Wedlock. We have no possible explanation for the differences observed between previous reports and those reported in the present work; however, both data reported in the literature were carried out in experimental transmission studies, in which the infection source was an intratracheal instillation of a virulent *M. bovis* strain [39,40]. To note, these differences in the experimental design could be related to the discrepancies observed in the results. Thus, field conditions trials have limitations but they must be considered as useful for showing the potential results of their use in real-world conditions. 

*M. bovis* BCG vaccination and its effect have been widely studied, even in a natural transmission setting, which could contribute to understanding several aspects related to vaccination. Thus, Lopez Valencia et al. [5] suggested that BCG vaccination could prevent the excretion of wild-type bacilli among cattle exposed to natural transmission of *M. bovis*, as implied by the absence of positivity in nasal swabs at 300 dpv. However, the authors were unable to confirm infection among the BCG-vaccinated animals due to the lack of availability to perform bacteriology [5]. These results contrast with those obtained in the present study, where *M. bovis* BCG-vaccinated animals showed higher positivity according to nasal swabs taken at the end of the trial (249 dprv) suggesting that vaccination may not prevent respiratory excretion.

In the present study, developed in a dairy farm managed under routine Argentinean husbandry conditions, the transmission rate of *M. bovis* wild-type strains was 23.1% (5.04–53.81), considering non-vaccinated animals. The observed transmission rate was lower than that reported by Ameni et al., as seen by LCT detection (85–86%; *p* = 0.001) and bacteriology (79–85%; *p* ≤ 0.001) [6,7]. However, Berggren et al. [37] reported a similar transmission rate to that observed in the present study for LCT (21%; *p* = 0.897) with an isolation rate that did not differ significantly (39%; *p* = 0.084) [36]. Additionally, the transmission rate obtained in this study was similar to that observed by Nugent et al. under extensive farming conditions (2.7%; *p* = 0.836) [9]. 

Considering the transmission rate observed in the present study, the exposition of the experimental groups to animals infected with *M. bovis* was not consistent with the planned 1:1 ratio, due to the management characteristics that were routinely performed in the dairy farm. This situation was confirmed at least once when a ratio of 0.5:1 was observed between naturally infected and tested (vaccinated and unvaccinated) animals. 

Information about necropsies of the naturally infected bTB cohort may have provided valuable information regarding the extent of LCT, and therefore, the likelihood of transmission, i.e., through respiratory excretion of the bacilli. However, farmers’ policies made necropsy of naturally infected cattle impossible. Other factors that may have influenced the likelihood of transmission of *M. bovis* wild-type strains, include differences in age ranges between the studied animals and those naturally infected, as well as the limited cohousing time. 

Although the transmission rate of the *M. bovis* wild-type strains was lower than expected, the non-vaccinated group had the lowest number of LCT, and despite certain limitations, all vaccines were equally affected by these factors, suggesting the low protection conferred.

Regarding excretion, 64.3% (35.1–87.2) of the animals vaccinated with the *M. bovis Δmce2-phoP* strain were positive in nasal swabs towards the end of the trial. This result is consistent with the macroscopic inspection and the bacteriology results that identified granulomas and viable *M. bovis* wild-type in the respiratory system. The *M. bovis Δmce2* and *M. bovis* BCGr groups showed similar percentages of positivity in nasal swabs (*p* > 0.45). This finding was also supported by the detected LCT and the positivity obtained by tissue-PCR in the respiratory system. This evidence suggests that vaccination with experimental vaccine candidates failed to prevent the excretion through the respiratory route. 

Despite detecting macroscopic/microscopic granulomas in all vaccinated groups, tissue-PCR and spoligotyping did not evidence the presence of genomic DNA corresponding to any of the experimental and control *M. bovis* vaccine strains. Molecular tools identified *M. bovis* strains belonging to the SB0131 spoligotype in all groups. This molecular type was different to the *M. bovis Δmce2/M. bovis Δmce2-phoP* (SB0267) and *M. bovis* BCG/*M. bovis* BCGr (SB0120) spoligotypes. 

This result indicates that the *M. bovis*-infections detected in the experimental groups were due to *M. bovis* wild-type strains. Previous research has also detected the presence of the SB0131 spoligotype in dairy bovines from herds in the region in which the trial was conducted [41]. Thus, the SB0131 spoligotype seems to be associated with naturally infected cattle. 

Transmission capacity of the experimental and control strains was an attribute addressed in the present trial. A previous study performed with the parental strain NCTC10772 of both experimental vaccines, demonstrated positivity in nasal swabs by PCR in cattle inoculated intratracheally [42]. Despite this previous report, in the present trial, experimental and control strains were not detectable in biological or environmental samples. This finding suggests that these experimental attenuated vaccine strains did not retain the capacity of transmissibility from the vaccinated bovines to other animals or to the environment through the respiratory or digestive route, when administered subcutaneously in the conditions assayed. 

An important limitation of using *M. bovis*-based vaccines is the interference with ante mortem diagnosis. Indeed, in the present study the non-vaccinated group exhibited CFT sensitization from 84 dprv with a low rate (15.4%, 95% CI: 1.5–45.4) which did not increase at the end of the trial, while IGRA detected an earlier sensitization at 42 dprv, with the same positivity rate as the CFT. At this latest time point, non-vaccinated bovines had cohabited with naturally infected bovines for 122 days. It should be noted that these were the first results consistent with sensitization in naïve animals within this study. This sensitization period agrees with periods previously obtained in trials conducted in Ethiopia and Chile, which reported the first evidence of sensitization of non-vaccinated animals at 120 days after exposure to positive reactor animals [6,10]. 

The reactivity of animals in the experimental groups, including non-vaccinated ones, was higher by IGRA (70.2%; 58.4–81.9) than by CFT (38.8%; 26.4–51.2). Infection with field strains did not have a stimulating effect, as evidenced by IGRA which showed positive results in vaccinated bovines both with and without LCT (83.3%; 51.6–97.9 and 64.3%; 48.6–69.9, respectively) (*p* = 0.368). In accordance with these findings, cattle with LCT had positive CFT results (41.7%; 15.2–71.3) similar to those obtained in vaccinated animals without LCT (42.9%; 26.7–59.0) (*p* = 0.796). Infection rates were calculated using a test that is not perfect in terms of sensitivity. However, this ante mortem tool is crucial for surveillance of bTB worldwide. For selection of naïve animals, the SCTST was used for the screening stage to ensure that the animals included in the study were negative for the intradermal reaction, taking into account that Argentina is not a bTB free country. However, during the trial, the CFT was chosen to study the immune response due to this technique being the official screening test routinely used and approved by the national program in Argentina [20].

Throughout the trial, IGRA reactivity fluctuated (some animals initially tested positive and later negative, and vice versa). Since vaccine strains have a reduced replication capacity and *M. bovis* BCG lacks genes encoding antigenic proteins, sensitization of vaccinated animals may result in temporary reactivity in ante mortem diagnostic tests. This temporal reaction has been documented as variable in *M. bovis* BCG vaccination trials lasting between 6 and 24 months [10,37,43,44,45]. 

Regarding sensitization, it is important to acknowledge that, given the very low reactivity observed after the initial immunization, the experimental design required an adjustment. This consisted of performing a re-vaccination while the animals had already been cohabiting with naturally infected bovines for more than 70 days. This aspect is critical because of the difficulty to determine whether the observed reactivity percentages are a consequence of the first immunization, exposure to natural infection, or both. 

The use of live attenuated strains as vaccine candidates against bTB requires the development of new diagnostic reagents that allow the differentiation between vaccinated and infected animals (DIVA diagnosis) [46]. This is particularly important owing to the interference of vaccination with diagnostic tests [13], as confirmed in vaccination trials conducted under experimental conditions with cattle vaccinated with *M. bovis Δmce2* and *M. bovis* BCG [18] or inoculated with virulent wild-type strains of *M. bovis* [42]. Furthermore, researchers have also confirmed this fact in natural conditions [6,7,9,10]. For example, a trial conducted in New Zealand revealed that a recombinant fusion protein containing ESAT6, CFP10 and EspC, used as an antigen for TST, could distinguish between TST sensitization due to vaccination with *M. bovis* BCG and a natural *M. bovis* infection [8]. 

In the present study, we detected sensitization of vaccinated animals with the antigen previously described by Srinivasan et al. [14]. The first vaccination period was crucial to evaluate the performance of this antigen since this was the phase in which the animals had not yet come into contact with naturally infected animals. However, as described above, the fact that in this period the vaccination failed to induce a cell-mediated immune response precluded conclusions regarding the antigenic performance of the FP reagent. Even though, both antigens, PPDB and FP, showed an increased tendency in the IGRA, showing higher positivity towards the end of the trial, no significant differences were observed between them. Furthermore, no definitive conclusion is attainable regarding the contribution of these antigens as DIVA reagent in the context of the non-*M. bovis* BCG vaccine strains used in the present trial. 

Regarding the mortality observed throughout the trial, the value (10%) was similar to those reported by Lopez-Valencia et al. and Abalos et al. in trials performed in natural transmission settings [5,10]. 

Despite not being statistically supported, the results obtained with *M. bovis ∆mce2-phoP* are different in absolute terms compared to the *M. bovis* BCG group. In developing countries, the goal of vaccinating cattle may be less strict and the main requirement is to reduce the spread of bovine tuberculosis. In the present study, no conclusive evidence was obtained that warranted promising results in further evaluation studies. However, the question remains to be answered whether the genetically modified *M. bovis ∆mce2-phoP* candidate, in other contexts, such as a trial encompassing vaccination and then exposure in a higher endemic herd compared to the one available in the present study, could achieve a real contribution to the prevention of bTB transmission. If so, other limitations need to be addressed, especially since it still requires a DIVA diagnosis.

## 5. Conclusions

Based on the results of this study, there is no statistical evidence to suggest improved protection of the vaccine candidates, including the well-characterized BCG. Re-vaccination of likely infected bovines did not highlight a conclusive result, even suggesting a detrimental effect on those vaccinated with *M. bovis BCG*. Despite these results, some questions remain to be answered related to the fact that the group vaccinated with *M. bovis ∆mce2-phoP* exhibited the lowest proportion of animals with macroscopic lesions, compared to *M. bovis* BCG; and the impact of vaccinating not infected animals. Should these issues be deciphered, other limitations must be addressed, especially given that it still requires a DIVA diagnosis. 

## Figures and Tables

**Figure 1 vaccines-12-01173-f001:**
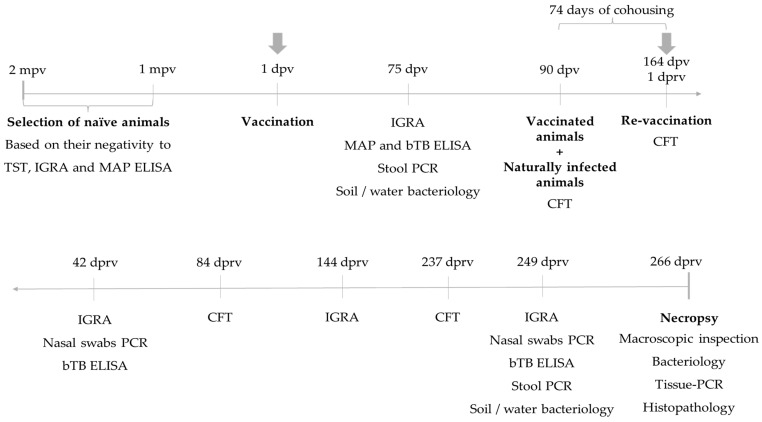
Timeline of the trial. A schematic timeline illustrating the most relevant intervention and sampling points, types of samples and techniques used to monitor the animals prior to the necropsy. mpv: months pre-vaccination. dpv: days post-vaccination. dprv: days post re-vaccination. TST: tuberculin skin test, IGRA: Interferon-Gamma release assay, MAP: *Mycobacterium avium* subsp. *Paratuberculosis*, ELISA: Enzyme-Linked Immunosorbent assay, CFT: caudal fold test, PCR: Polymerase Chain Reaction, bTB: bovine tuberculosis.

**Figure 2 vaccines-12-01173-f002:**
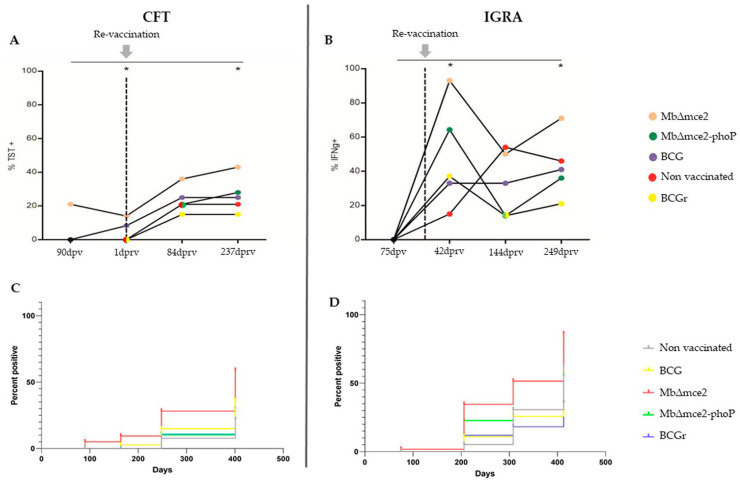
(**A**). Prevalence of animals positive to the caudal fold test (CFT) in each group at the different sampling times. (**B**). Percentage of positive animals to the interferon-gamma release assay (IGRA). Kruskal–Wallis test and Dunn’s post test. * Prevalence that differed significantly between the studied groups, *p* < 0.05. The dotted vertical line and gray arrow indicate the day of the re-vaccination. (**C**,**D**). Incidence of positivity for both CFT and IGRA, respectively, represented by the Kaplan–Meier analysis. TST: Tuberculin skin test. Dpv: days post vaccination. Dprv: days post re-vaccination. IFNg: Interferon gamma.

**Figure 3 vaccines-12-01173-f003:**
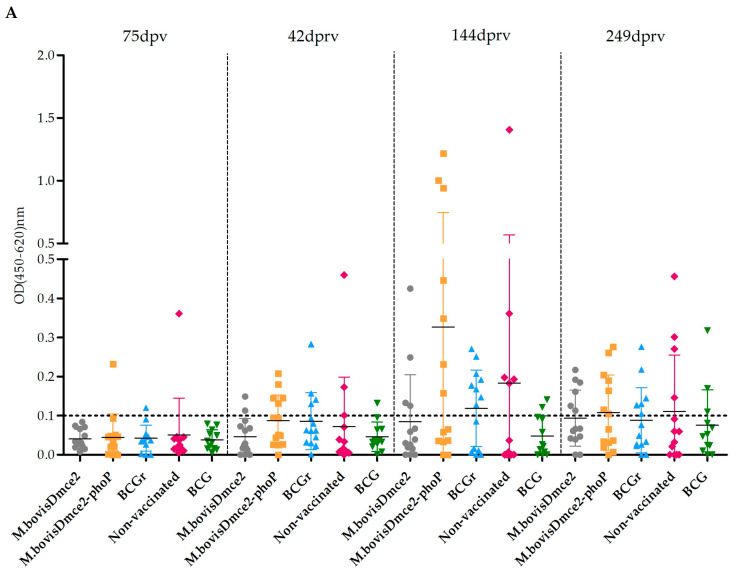
Scatter plot showing the OD readouts for Interferon Gamma Release Assay (IGRA) detected in cattle from the different groups under study at 75 days post vaccination (dpv), 42, 144 and 249 days post re-vaccination (dprv) when stimulated with PPDA (**A**), PPDB (**B**) and FP (**C**). The dotted horizontal line represents the cut-off of 0.1, above which is considered a positive OD value and below which is negative. Kruskal–Wallis Test and Dunn’s post test. *, *p* < 0.05; **, *p* < 0.1; *** *p* < 0.001.

**Figure 4 vaccines-12-01173-f004:**
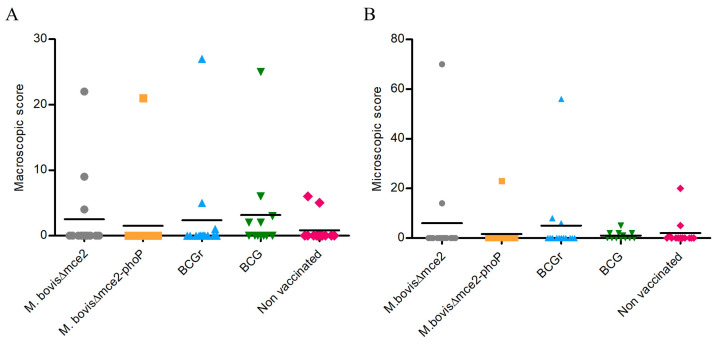
(**A**). Macroscopic lesion score. (**B**). Microscopic lesion score of each animal from the different groups.

**Table 1 vaccines-12-01173-t001:** PCRs used in the present study to identify *M. bovis* genome.

Name	PCR Product (bp) *	Sample	Bibliography
β actin-PCR	99	Tissue DNA	[27]
IS*6110*-PCR	246	Tissue DNA/*M. bovis* culture lysate	[28]
Rv*2807*-PCR	443	Tissue DNA/*M. bovis* culture lysate	[29]
Mut *Mce2* del-PCR	177 (absence) 1831 (presence)	Tissue DNA/*M. bovis* culture lysate	[18]
*PhoP*-PCR	322	Tissue DNA/*M. bovis* culture lysate	[19]
*esxA*-PCR	288	Tissue DNA/*M. bovis* culture lysate	[30]
*esxB*-PCR	450	Tissue DNA/*M. bovis* culture lysate	[30]

* bp: base pairs.

**Table 2 vaccines-12-01173-t002:** Macroscopic and microscopic score of lesions.

Strain	Macroscopic Lesions ^1^	Score LNs ^2^	Score L ^2^	Microscopic Lesions ^1^	Score LNs ^3^	Score L ^3^
Non-Vaccinated	2/13	11	3	3/13	15	0
*M. bovis* BCG	5/12	40	0	4/12	15	0
*M. bovis* BCG*r*	3/14	28	7	3/14	61	7
*M. bovis Δmce2*	3/14	17	18	2/14	36	28
*M. bovisΔmce2*-*phoP*	1/14	16	9	1/14	18	3

^1^ Number of animals from the total with macroscopic and microscopic lesions identified during Slaughter inspection and necropsy for each study group and pathology score. ^2^ Macroscopic score in lymph nodes (LNs) and lung (L). ^3^ Microscopic score in LNs and L.

## Data Availability

Research data will be available upon request to corresponding authors.

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
