# Peer review of "Field Trial with Vaccine Candidates Against Bovine Tuberculosis Among Likely Infected Cattle in a Natural Transmission Setting"

_vaccines, 2024, doi:10.3390/vaccines12101173_

Round 1

Reviewer 1 Report

Comments and Suggestions for Authors

The authors present an important study concept - to address the potential of vaccine candidates to prevent transmission of M. bovis between cattle. The study uses a natural transmission model where animals detected using established TB diagnostic tests were in contact with animals vaccinated with a range of vaccines, including BCG. The presence of M. bovis was assessed in a range of samples, and at the end of the study lesions typical of TB were measured in the tissues of the respiratory tract. This type of study design has previously been used to demonstrate significant ability of BCG to reduce TB transmission in natural settings. 

There are a number of very significant limitations of this study which the authors acknowledge to some extent. In particular, there was no evidence at all that the original vaccination induced an immune response - this seems very strange and no explanation is offered. No kinetic analysis of the immune response induced by vaccination was carried out prior to exposing the animals to the naturally infected TB cohort - this is a significant oversight as effectively at the first point of contact with M. bovis all of the animals were effectively non-immune and therefore differences in the groups are very difficult to interpret. By the time re-vaccination was given, the animals had been exposed to the naturally infected cattle for 70+ days - therefore revaccination could have been given to already infected cattle: the impact of this is likely to significantly confound any interpretation of the data. 

Significant evidence of TB lesions was presented for all of the groups, with many microscopic lesions being detected in vaccinated animals. Does this suggest an alteration in the immune response that could be associated with 'protective/immune' lesions controlling bacteria? No further studies were done to assess this and at face value the data simply show that (a) vaccination was not successful and (b) re-vaccination had no impact or was even detrimental.

The paper has some merit and it is clearly a large, longitudinal, valuable study in some respects. However, it should undergo a substantial rewrite, be shortened and make additional comment on the impact of vaccinating cattle that are already likely infected (this will be important if large scale vaccination is rolled out to the cattle industry) and what the difference in macroscopic/microscopic lesions could mean in terms of protective immunity.

Comments on the Quality of English Language

Some corrections are needed for typographical errors.

Reviewer 2 Report

Comments and Suggestions for Authors

The study is of interest as it evaluates vaccines that may improve the protection conferred by BCG in cattle. Currently, there are not many alternatives to BCG under study, so the results are important, especially since field conditions have been applied to draw conclusions. My main concern is regarding the conclusions drawn by the authors, who emphasize the need for further studies with one of the candidates due to promising results. In my opinion, the data provided are enlightening regarding the low protection conferred by the vaccine candidates, at least in terms of not improving upon other previously evaluated alternatives, including BCG or even inactivated vaccines, which dampens optimism about them. Even considering some of the study's limitations, which are acknowledged by the authors, there are no indications suggesting that further evaluation of these vaccines in other contexts is warranted. The field conditions employed have limitations, but they are useful for showing the potential results of their use in real-world conditions and also highlight the limitations of experimental models in demonstrating efficacy. In my opinion, these are the aspects that should be included in the discussion, rather than raising hope for a vaccine that, based on the results, does not seem to offer much improvement over existing options, especially given that it is genetically modified and still requires a DIVA diagnosis. Another aspect to improve is related to grammar, which needs to be reviewed. The title of the paper could be modified to be more descriptive, as it currently seems more like the title of a literature review rather than an experimental study. The paper contains numerous figures and tables, and not all of them are strictly necessary or could be improved. Figures 1 and 2 have small text and are barely readable. Figure 4 should be presented in a better format to facilitate comparison between vaccines. I believe Figure and Table 5 are not strictly necessary, and the results can be described in the text or added as supplementary material.

Other points:

Abstract: revise the grammar, some sentences can be improved for a better understanding (lines 36-40). Line 44: lower? Line 43-44: This does not seem to agree with what is indicated in lines 460-462 and 469-470, please clarify.

Lines 49-50: I prefer to describe the bovine TB as the disease in bovines (cattle). Bovine TB (cattle) is subjected to eradication campaigns so if it is referred to M. bovis infection in animals and humans has no sense. Bovine TB can be caused by M. bovis and in a lesser extent by other MTBC members.

Line 78: I guess SCITST is not used again, perhaps it is not neccessary.

Line 112: pathogenicity conferred? perhaps better protection in terms of reducing pathogenicity or similar

Line 124: the use of CFT is relevant since sensitivity/specificity differ from that obtained using the cervical test and therefore it can affect to the results (infection rates). This is is something relevant for discussion.

Line 129: M. avium subsp. paratuberculosis

Line 145: three

Line 150: bTB?

Line 188: CFT?

Line 226: bovine tuberculin PPD... can be removed since it has been used in line 196.

Lines 370-383: this information is not statistical analysis and should be included in the previous sections.

Line 393: the term incidence can be confusing since it is not based on infection status (e.g. bacteriology), add apparent or use CFT reactivity.

Lines 403-409: it is no clear since results are based on the using of PPD or the fusion protein. Please clarify.

Lines 460-461.: in my opinion, this result is a key point to modify the abstract and conclusions.

Lines 460-476. if finally the differences were not significant, has no sense to highlight the highest or lowest scores in order to extract conclusions.

Line 478: remove "and"

Table 4. included the totals for each group for a better understanding or %.

Line 508-511: Can I interpret then that using BCG the risk of infection was the lowest?Can be affected by a different reactivity to the CFT depending of the vaccine?

Line 513: TB

Lines 534-539: I do not agree with this. The authors already state earlier that the non-vaccinated group had the fewest lesions, and despite certain limitations, all vaccines were equally affected by these factors, suggesting the low protection conferred.

Lines 555-558: In my opinion , the results are far of be promising. The differences were not significant.The lesions were not significantly lower and the excretion was also similar.

Line 639: perhaps the authors can provide differences in terms of reactivity (sensitivity) between PPD and fusion protein at the end of the study.

Lines 646-647: Attending to the results, I do not agree with this statement.

Lines 648-656: to highlight the limitations of the study is important but the should be integrated throughout the discussion, the last paragraph should be focused in the main conclusions. The natural infection models have also advantages in comparison to the experimental ones and can reflect better the field conditions. Asa I mentioned before, rates of infection were calculated using a test that is not perfect in terms of sensitivity.

Comments on the Quality of English Language

English grammar should be revised throughout the manuscript to improve the clarity of some sentences.

Round 2

Reviewer 1 Report

Comments and Suggestions for Authors

The authors have addressed the comments. There is a requirement for editing of English language, predominantly in sentence construction.

Comments on the Quality of English Language

Some editing of sentence construction is needed.

Author Response

Dear Reviewer, the manuscript was impoved regarding english language and conclusion. 

 Sincerely yours, 

MEE, corresponding author